# Carboxyl Functionalization of N-MWCNTs with Stone–Wales Defects and Possibility of HIF-1α Wave-Diffusive Delivery

**DOI:** 10.3390/ijms24021296

**Published:** 2023-01-09

**Authors:** Vladislav V. Shunaev, Nadezhda G. Bobenko, Petr M. Korusenko, Valeriy E. Egorushkin, Olga E. Glukhova

**Affiliations:** 1Department of Physics, Saratov State University, 410012 Saratov, Russia; 2Laboratory of Physics of Nonlinear Media, Institute of Strength Physics and Materials Science of SB RAS, 2/4 Academichesky Avenue, 634021 Tomsk, Russia; 3Department of Physics, Omsk State Technical University, 11 Mira Prosp., 644050 Omsk, Russia; 4Institute for Bionic Technologies and Engineering, Sechenov University, 119991 Moscow, Russia

**Keywords:** multi-walled carbon nanotubes, Stone–Wales defects, HIF-1α, molecular modeling, drug delivery, hybridization

## Abstract

Nitrogen-doped multi-walled carbon nanotubes (N-MWCNTs) are widely used for drug delivery. One of the main challenges is to clarify their interaction with hypoxia-inducible factor 1 alpha (HIF-1α), the lack of which leads to oncological and cardiovascular diseases. In the presented study, N-MWCNTs were synthesized by catalytic chemical vapor deposition and irradiated with argon ions. Their chemical state, local structure, interfaces, Stone–Wales defects, and doping with nitrogen were analyzed by high resolution transmission electron microscopy (HRTEM), Raman spectroscopy, X-ray photoelectron spectroscopy (XPS), and near-edge X-ray absorption fine structure (NEXAFS) spectroscopy. Using experimental data, supercells of functionalized N-MWCNTs with an oxygen content of 2.7, 4 and 6 at. % in carboxyl groups were built by quantum chemical methods. Our analysis by the self-consistent charge density functional tight-binding (SCC DFTB) method shows that a key role in the functionalization of CNTs with carboxyl groups belongs to Stone–Wales defects. The results of research in the decoration of CNTs with HIF-1α demonstrate the possibility of wave-diffusion drug delivery. The nature of hybridization and relaxation determines the mechanism of oxygen regulation with HIF-1α molecules, namely, by OH-(OH–C) and OH-(O=C) chemical bonds. The concentration dependence of drug release in the diffusion mode suggests that the best pattern for drug delivery is provided by the tube with a carboxylic oxygen content of 6 at. %.

## 1. Introduction

Currently, single- and multi-walled carbon nanotubes (SWCNTs, MWCNTs) are widely used in biomedicine for clinical diagnostics (electroencephalography, electrocardiography, electromyography) and for tissue engineering [1,2,3,4]. The use of defect-free CNTs is limited by their weak solubility, short half-life, immunogenicity, toxicity, etc. [5], which can be obviated through functionalization [6,7,8].

Such functionalized CNTs can serve for pharmacology, diagnostics, and treatment of different diseases, including oncological examples, the transfer of proteins and genes through cell membranes to nuclei, and regulation of their expression and inhibition [9,10,11].

The role of functionalized CNTs in biological processes can vary with their type. Of importance is their role in the behavior of hypoxia-inducible factor 1 alpha (HIF-1α), which is a trigger of fight against different diseases [12,13]. If the HIF-1α content is low, it suppresses the immune function, cartilaginous tissue formation, wound healing, etc. [13,14,15,16]. If the HIF-1α content is high, it points to oncological or cardiovascular diseases: hypoxia, strokes, heart attacks, pulmonary hypertension, etc. [10,11,12,13,14,15], which disturb, along with concomitant diseases, the self-regulation of HIF-1α and its stabilization [13]. Therefore, increasing or decreasing the HIF-1α content through pharmacological intervention can help to treat many types of diseases [12,13,14,15]. The concentration of a drug released from its carrier varies linearly or exponentially with time, which is normally described by the diffusion equation [16,17]. However, as has been found experimentally, wavelike kinetics of drug release are also possible [18]. The mechanism of wave-diffusion release is currently unclear.

Going back to HIF-1α and to the effect of functionalized CNTs, it should be noted that the addition of siRNA-SWCNTs greatly inhibits the cellular activity of HIF-1α, and the addition of Oxygen-SWCNTs decreases its expression [10,11]. Increasing the concentration of oxygen delivered to cells by SWCNTs also increases the number of proteins that, being controlled by HIF-1α molecules, participate in apoptosis, autophagy, survival, and growth of cells [10]. Thus, functionalized CNTs are an effective drug carrier for the treatment of different diseases [4,5,10,11].

Among such effective drug carriers are CNTs functionalized with carboxyl groups (COOH); the delivery of drugs with COOH-CNTs is analyzed elsewhere [19,20,21,22]. This type of drug carrier provides covalent and noncovalent bonds between drug molecules and COOH [20,21,22] and thermodynamically beneficial exothermal drug adsorption on its surface [19,20,22]. The possibility of the formation of a drug–COOH structure and its stability are determined by the minimum binding energy. The microscopic mechanism that provides such a structure is not quite clear. Because its experimental studies cannot be thorough, quantum methods are applied to explore the possibilities of the attachment of different functional groups and drug biomolecules to CNTs, allowing estimations of quantum molecular descriptors, free energies of salvation, binding energies [19,20,21,22,23,24], etc.

The results of density functional theory (DFT) research in the functionalization of CNTs with carboxyl groups and its effect on the electronic properties [23,24] show that the binding energy and the electron transfer between CNTs and carboxyl groups reduce with the increasing CNT diameter [23] and that the electronic characteristics of CNTs strongly depend on the concentration and location of carboxyl groups [23,24]. All the studies cited above consider the attachment of carboxyl groups to defect-free CNTs.

However, actual MWCNTs functionalized with carboxyl groups contain different types of defects, which arise at their interfaces during fabrication, nitriding, irradiation, etc. [25,26,27]. In addition, COOH-N-MWCNTs are biocompatible with tissues, nontoxic (unlike pure CNTs), and capable of attaching protein and drug molecules; that is, they have a high potential for pharmacological use [10].

The aim of our study is to trace the variation in the atomic and electronic structure of N-MWCNTs, to analyze the mechanism of their carboxyl functionalization and the behavior of HIF-1α during its interaction with COOH-N-MWCNTs having different oxygen concentrations in COOH groups, and to clarify and describe the nature of wavelike drug delivery on the example of HIF-1α-COOH-N-MWCNTs.

## 2. Results and Discussion

### 2.1. High-Resolution Transmission Electron Microscopy (HRTEM)

Figure 1a shows an HRTEM image of the irradiated N-MWCNT side wall having a near-cylindrical bamboo structure with an average outer diameter of ~20 nm. As can be seen, the outer side of the tube reveals a large number of Stone–Wales defects (arrows) compared to the initial N-MWCNT (see Appendix A) [28].

### 2.2. Raman Spectroscopy

Figure 1b shows the spectrum of Raman scattering by the irradiated N-MWCNTs at 800–1800 cm^−1^. The spectrum contains two overlapped peaks: G at ~1590 cm^−1^ and D at ~1390 cm^−1^. Its expansion in Lorentzian functions shows that the peak G is represented by only one function corresponding to a zigzag structure [29], and the peak D is split into three: D1, D3, and D4. According to [30], D1 corresponds to topological defects, D3 to amorphous-like carbon, and D4 to impurities, dangling bonds, and oxygen groups.

The positions of the maxima of the Lorentzian functions and the ratio of the integral peak intensities I_D_/I_G_, I_D1_/I_D_, I_D3_/I_D_, and I_D4_/I_D_ are tabulated in Table 1. The ratio I_Di_/I_D_ (i = 1, 3, 4) gives the relative content of the above defects: 0.7(D1)/0.2(D3)/0.1(D4), suggesting that the disordering of the N-MWCNT structure is contributed mostly by topological defects (~70%).

The crystallite size was calculated by the formula $L_{a}=(2.4*10^{-10})\lambda^{4}\frac{I_{D}}{I_{G}}$ [31]. Its value in the irradiated N-MWCNTs measures 5.6 nm, which is much smaller than both *L_a_* = 12 nm in the non-irradiated N-MWCNTs and *L_a_* = 24 nm in the MWCNTs [25,32].

Thus, carbon nanotubes with nitrogen embedded in their walls (N-MWCNTs) and N-MWCNTs after ion beam irradiation have a significantly smaller crystallite size than MWCNT without nitrogen. Such changes in the structure of MWCNTs lead to an increase in the density of interfaces and, consequently, in the number of topological defects formed in interface regions.. These defects are mainly Stone–Wales defects [33,34], which is confirmed by our TEM research.

### 2.3. XPS (X-ray Photoelectron Spectroscopy)

Figure 1c shows the survey photoemission (PE) spectrum of the irradiated N-MWCNTs. As can be seen, the spectrum contains a C1s carbon line at ~285 eV, a N1s nitrogen line at ~401 eV, an O1s oxygen line at ~533 eV, and oxygen and carbon KLL Auger transition lines at ~343 eV and ~590 eV, respectively. The most intense among these lines are those of C1s and O1s, which shows that these elements dominate in the nanotube structure. The atomic concentration calculated from the survey PE spectrum is 85.22 at. % for carbon, 1.87 at. % for nitrogen, and 12.91 at. % for oxygen (see Appendix A).

The N1s PE spectrum of the irradiated N-MWCNTs was fitted by the Gaussian–Lorentzian product formula (40:60) with two components against five components in the initial nanotubes (Figure 1d and Appendix A). The component with a peak at ~401 eV corresponds to substitutional nitrogen (N1), and the high-energy component with a peak at ~403.5 eV corresponds to oxidized nitrogen (N2), primarily to pyridine nitrogen. From our quantitative data, the N1 and N2 concentrations measure 1.71 at. % and 0.16 at. %, respectively; that is, the major contribution is provided by substitutional nitrogen after ion beam irradiation.

The C1s PE spectrum of the irradiated N-MWCNTs is well approximated by five components (Figure 1e). The most intense spectral component with a peak at a binding energy of 284.8 eV corresponds to the carbon bonds C=C (sp^2^), and the component with a peak at 285.4 eV corresponds to the bond C–C (sp^3^), which is modified by structural defects, and to the bond C–N [25]. The component with a peak at ~287 eV is due to the bonds C–O, C–O–C, and C–OH [32]. The peaks at 289 eV and 290 eV are due to the bonds of carbon and oxygen in carbonyl (C=O) and carboxyl (COOH) groups, respectively [32]. The relative areas of carbon components (%) in the total sum of these bonds is 46.1% for C=C, 31% for C–C and C–N, 12.9% for C–OH, 6.1% for C=O, and 3.9% for O=C–OH. Our analysis of the O1s spectrum (not shown here) shows that the amount of oxygen in oxygen-containing groups is 6 at. % in C–OH, 3 at. % in C=O, and 4 at. % in O=C–OH. The oxygen concentrations in these groups are comparable.

### 2.4. Near-Edge X-ray Absorption Fine Structure (NEXAFS)

Figure 1f shows the C1s NEXAFS spectrum of the irradiated N-MWCNTs formed by absorption transitions of electrons from the 1s state of carbon to unoccupied 2p π* and σ* states due to X-ray quantum absorption at a photon energy of 280–320 eV. The presence of intense absorption bands π*(C=C) at a photon energy of ~284.9 eV and σ*(C=C) at ~291–293 eV in the C1s NEXAFS spectrum shows that the N-MWCNT contains regions with a hexagonal structure [35,36]. However, the absence of local peaks at ~291.7 and ~292.8 eV, which should be present in perfectly structured MWCNTs, and the smearing of the high-energy spectrum of σ*(C=C) point to a large number of structural defects in the specimen [37,38]. The peaks of π*(C–OH) at a photon energy of ∼286.4 eV and π*(C=O) at ∼288.5 eV correspond to carbon in hydroxyl, carbonyl, and carboxyl groups [35,39]. The above spectral features suggest that oxygen-containing functional groups are present on the CNT surface.

Our experiments show that the CNT specimens have a mainly zigzag bamboo structure with a high crystallite interface density, crystallite size *L_a_* = 5.6 nm, and two-percent substitutional nitrogen content. The location of defects, which are mostly topological Stone–Wales defects, falls on the region of interfaces. At the Stone–Wales defects of the interface region, the nanotube surface is functionalized with C–OH, C=O, and O=C–OH groups in comparable concentration proportions. Below we focus mainly on the carboxyl group because it is this group that is of importance in different medical problems [19,20,21,22].

### 2.5. DFT Calculation of Carboxyl Functionalization and Its Enhancement with Stone–Wales Defects

In the N-MWCNT specimen under study, the oxygen content in all the groups listed above is 12.9 at. %, of which 4 at. % falls on the carboxyl group. At an oxygen concentration of 2.5–5.5 at. %, the carboxyl group forms short-range ordered structures, and at ≥6% its attachment to interfaces results in long-range ordered structures [40]. Below we analyze the carboxyl functionalization of N-MWCNTs in view of the two types of structure.

Let us start with DFT calculations of the electronic structure and parameters of chemical bonds for the N-MWCNTs with an oxygen content of 2.7, 4, and 6 at. % in their carboxyl group, with Stone–Wales defects, and with substitutional nitrogen and carbonyl group in their surface layer. Hereinafter, the structures with an oxygen content of 2.7, 4, and 6 at. % are denoted as S1, S2, and S3, respectively.

The nanotube diameter at the sites of carboxyl and HIF-1α attachment is ~20 nm, and, therefore, these sites can be taken as the plane. Because carboxyl groups at oxygen concentrations of 2.7% and 4 at. % form short-range order structures at nanotube interfaces, the nanotube interfaces at such concentrations can be modeled by finite graphene-like cells with rigidly fixed edges (Figure 2a,b). In our model, the cell length was equal to 17 hexagons, and the cell width to 3.5 hexagons. Each cell contained three Stone–Wales defects with substitutional nitrogen atoms and oxygen C=O. On the S1 cell surface, there were two carboxyl groups O=C–OH (two oxygen atoms per 74 atoms in a cell), and on the S2 cell surface there were three carboxyl groups (two oxygen atoms per 51 atoms in a cell). The S3 structure formed by carboxyl groups has a long-range order. Its cell was modeled by 35 atoms with one carboxyl group (black frame in Figure 2c) such that it can be translated in two directions. In this structure, the oxygen concentration in O=C–OH reaches 6 at. % at a total atomic concentration of N, O, and H greater than 15%. The concentration of these atoms in a cell is 3 at. % for nitrogen (one N atom per 35 atoms), 9 at. % for oxygen (three O atoms per 35 atoms), and 3 at. % for hydrogen (one H atom per 35 atoms), which makes up 15 at. % in total.

Let us consider the electronic characteristics calculated for the three structures. The density of electronic states (DOS) and the partial density of electronic states (PDOS) for each structure are presented in Figure 3. As can be seen, the Fermi levels for S1 (Figure 3a) and S2 (Figure 3b) are almost coincident: −5.57 and −5.60 eV. In both structures, the highest DOS is observed at the Fermi level, which points to their metallic conductivity and high chemical activity. In the S1 structure, peak 1 falls on −5.35 eV, corresponding only to oxygen, and peak 2 (O and OH) on −6.05 eV; the most intense peak belongs to oxygen. The energies of peaks 3, 4, and 5 are −7.20 eV, −8.49 eV, and −9.77 eV, respectively, and are the same for C, O, and OH. In the S2 structure, peak 1 (only O) falls on −6.11 eV, and peak 2 (O and OH) on −6.73 eV. The energies of peaks 3, 4, and 5 (C, O, and OH) are −8.02 eV, −9.38 eV, and 9.87 eV, respectively, and are the same for C, O, and OH as in the previous case.

The S3 structure (Figure 3c) differs greatly in DOS and PDOS from S1 and S2. Here, the lowest DOS is observed at the Fermi level with an energy of −4.48 eV, which is close to the Fermi level in the initial nanotube (−4.70 eV). The energies of peaks 2, 3, 4, and 5 for C, O, and OH coincide and fall on −5.77, −7.22, −8.47, and −9.90 eV. Peak 1 is absent.

Varying the oxygen concentration hardly changes the binding energy *E_b_* between OOH and the nanotube surface. The Fermi level, DOS at the Fermi level, and binding energy for the nanotubes with S1, S2, and S3 structures are presented in Table 2.

Our calculations show that the carboxyl concentration on the N-MWCNT surface hardly affects the interatomic distances and the charge flow in the carboxyl group (Table 2). The interatomic distance for C=O is equal to 1.23 Å and for C–OH to 1.36 Å. The adsorption distance for C–C measures 1.505 Å, which differs from its earlier estimate (1.56–1.58 Å) [23,24,41,42].

The charge −0.68 e released by carbon from the carboxyl group is distributed between O (+0.55 e), OH (+0.11 e), and the nearest C in the surface layer (+0.02) (Table 2). The lack of charge on the COOH group is 0.02 e, which agrees with data reported elsewhere [23]. According to [24], the excess charge on COOH is 0.04–−0.13 e. In the latter study, the charge redistribution provides values greater than those obtained elsewhere [23] and gives −0.18 e on C, 0.22 e on O, and −0.05 e on OH.

It should be emphasized that the diminution of energy is a mere motive force, rather than a cause or a mechanism of binding. The binding mechanism provides the way to distinguish a desirable state.

Let us consider the mechanism by which carboxyl is bound to a nanotube. The formation of O=C–OH on the nanotube surface requires overcoming the energy barrier associated with their repulsion. To estimate the energy barrier, the repulsion energy *E_rep_* versus the interatomic distance C=O was plotted (Figure 4) for which O and OH were removed from a C atom with a step of 0.1 Å to 5.52 Å perpendicular to the tube surface. At each step, the structure was geometrically relaxed. The red curve in Figure 4 was calculated for the case in which no Stone–Wales defects were present and the C atom was rigidly fixed by other carbon bonds. The coordinates of this atom of the carboxyl group were such that the coordinates of its equilibrium position were always constant. It is seen from the curve that the difference in *E_rep_* between its initial and final position is 3.7 eV, which is just the desired potential barrier. The black curve in Figure 4 corresponds to the presence of Stone–Wales defects and weaker bonds of the C atom with N and O. This allows the C atom and its surroundings to change their positions. Numerals 0, 1, 2, and 3 on the black curve indicate the points of the most characteristic changes of atomic positions and repulsion energy *E_rep_*. In this case, the initial position of the O atom (point 0) corresponds to the C atom position in the surface plane, which determines the higher value of *E_rep_* compared to the previous case. As the O and OH groups approach the surface, the repulsion energy of the system increases almost linearly. Point 1 on the curve corresponds to the first local peak. In this position, the oxygen atom from the tube surface comes up above it and gets close to free oxygen (Figure 5a, inset). As the distance between the atoms and the surface at point 2 decreases, the repulsion energy drops steeply. This is due to the formation of a C=O bond, actually a carbonyl group, with the OH– group being unattached (Figure 5b, inset). The carboxyl group is formed from the carbonyl group through its polarization by residual OH–, charge transfer (Figure 4, inset), bond rupture between carbon and nitrogen, and attachment of OH to this carbon. Point 3 corresponds to the equilibrium position of the carboxyl group on the tube (Figure 5, inset). For this point, the energy barrier (the difference between the initial and final positions on the black curve) is a mere 1.62 eV, which is much lower than its value in the case of rigid C fixation. Such a decrease in the energy barrier is owed to the presence of Stone–Wales defects.

To determine the parameters of chemical bonds in the presence of Stone–Wales defects, let us consider the behavior of PDOS in the carboxyl group on approaching the tube surface. It is seen in Figure 5a that the PDOS peaks of O and OH at a C=O distance of 4.73 Å (Figure 4, point 1) fall on the same energy. All PDOS peaks are presented in Table 3: peak 1 at −4.08 eV, peak 2 at −4.92 eV (near the Fermi level), and peaks 3 and 4 at −5.35 eV and −6.00 eV, respectively. Table 3 also presents the variation in the charge *q* transferred between O, OH, and C on approaching the tube surface. In addition to the above peaks, there is one more peak (peak 5) corresponding to OH at −7.73 eV. The only separate peak of the C atom falls on −6.12 eV. At point 2, after optimization of the structure, a C=O carbonyl group is formed, changing the atomic positions and the behavior of PDOS. Peak 1 for O and OH (Figure 5b) shifts closer to the Fermi level (−4.67 eV). Peak 2 for OH alone, peak 3 for O alone, and peak 4 shift toward to the lower energy range. In the equilibrium position, corresponding to the carboxyl formation (Figure 5c), peak 1 disappears, and peaks 2, 3, and 4 shift more substantially and fall on the same energies for C, O, and OH. This means that the formation of a carboxyl group requires that the electronic states of bonds be close in energy, which also confirmed the previous DTF calculations [23].

The calculated electronic structure has the following features. The density of states of non-carboxyl carbon on the Fermi level at a concentration of 2.7 at. % and 4 at. % is represented by an intense peak, and its value on the Fermi level at 6 at. % is minimum (Table 2). The PDOS values of all carboxyl elements are close in energy. This means that the attachment of the functional group requires that the electronic states of interacting atoms be close in energy.

Thus, the substantial binding energy of the carboxyl group with the tube and the charge transfer from the tube to carboxyl oxygen are indicative of high energies of carboxyl and tube hybridization. In this case, the energy positions of PDOS peaks corresponding to carboxyl and tube electrons are coincident.

The presence of Stone–Wales defects decreases the energy barrier to the formation of O=C–OH bonds by weakening the bonding of carboxyl carbon and its nearest neighbors with the rest of the carbon tube surface. An important role in the attachment of OH to carbon belongs to nitrogen atoms as these atoms allow carbon to break the C–N bond for the formation of carboxyl.

The intermediate group for carboxyl–carbonyl corresponds to the minimum repulsion potential. The presence of carbonyl groups agrees with our experimental data (XPS, NEXAFS). The formation of carboxyl from carbonyl follows the rupture of carbon bond with nitrogen and the attachment of OH to this carbon. Without OH, the functionalization would be carbonylic. However, OH– polarizes the carbonyl group via charge transfer (here, the charge transferred on OH is maximal (Figure 4, inset)) and forms O=C–OH through its attachment.

### 2.6. Hybridization and Wave-Diffusion Drug Delivery

At the next stage of our study, molecules of HIF-1α protein with a structure were positioned in the S1, S2, and S3 cells above the carboxyl groups [43]. The most beneficial sites of protein location were determined in numerical experiments (Figure 6). The cells corresponding to the regions of interfaces had one attached molecule in the S1 structure and two molecules in the S2 one. In the S3 structure, one protein molecule was attached to a cell obtained by translating three unit cells with 35 atoms each. Thus, the protein amount per cell in S3 is smaller than in S2. However, compared to S1 and S2, the S3 structure has a much higher interface density, and hence the number of protein molecules on the whole tube with such a structure is much larger.

In S1 and S2, the protein molecules are bound to the OH group of carboxyl by the OH group. In S1, there is one stable structure with a bond length of 1.603 Å (Figure 6a). In S2, two variants of protein location are stable (Figure 6b). In the first variant, the molecule is located like in the previous structure, and in the second one, it is oriented, as shown at the right of Figure 6b. In the second case, the bond length is equal to 1.896 Å.

In S3, there are two OH bonds of the protein with carboxyl elements: with OH and with O (Figure 6c). The bond length is 1.876 Å for OH–OH and 1.936 Å for OH–O.

The binding energy of HIF-1α and COON-N-MWCNT for S3 is lowest (−0.64 eV), which is explained by the presence of two chemical bonds and boundary conditions. The binding energy for S1 with a bond length of 1.603 Å is −0.51 eV. For S2 with the same bond length, the binding energy is also −0.51 eV, and with 1.896 Å it is −0.35 eV. In all cases, the main type of protein interaction with the functionalized N-MWCNT is non-covalent hydrogen bonding, as is the case in SWCNTs [19].

Figure 6d–i shows the DOS and PDOS of the COOH-N-MWCNT with HIF-1α at different carboxyl concentrations. After the addition of HIF-1α, the Fermi level for S1 and S2 remains almost unchanged, whereas this level for S3 shifts from −4.48 to −4.73 eV. In all three cases, the HIF-1α states are separated by an energy gap of ~3 eV against the background of a high total density of states. The energy positions of the PDOS peaks for HIF-1α and nanotube are non-coincident, which points to their weak hybridization. This is also confirmed by the PDOS of the OH group of the tube and OH of the protein with hydrogen bonding. In S1 and S2, peaks 1 and 3 in the OH zone of the protein are positioned at much lower energies compared to OH peaks 2 and 4 of the tube. In S3, an additional O peak (peak 4), which is absent in S1 and S2, appears due to the second protein OH bond with carboxyl oxygen. Peak 1 goes above the Fermi level.

The last column of Table 4 presents the energy difference ω between peaks 1 and 2 for S1 and S2 and between peaks 1 and 2, as well as 3 and 4, for S3.

Thus, the binding energy *E_b_* between HIF-1α and COOH-N-MWCNT is an order of magnitude lower than *E_b_* between COOH and N-MWCNT, suggesting that the bonds between the former are non-covalent. The charge transfer between COOH and HIF-1α is also an order of magnitude lower than that between the carboxyl group and the tube. The key role in the charge transfer belongs to nitrogen (Table 5). The foregoing along with non-coincident positions of the PDOS peaks of the groups involved in binding suggests that the hybridization energy of HIF-1α and COOH-N-MWCNT is low. The lowest binding energy *E_b_* and two chemical bonds in S3 provides the most stable HIF-1α-COOH-NMWCNT structure among S1, S2, and S3.

Let us analyze the charge transfer between carboxyl and protein. As follows from Figure 7, carboxyl OH gives up 0.01 e to protein OH. Carboxyl oxygen takes up the rest of the charge transferred on 0.01 e: from carboxyl C, OH, and the nearest protein C, N, and H donor atoms indicated in Figure 7. Note that the charge is transferred not by the nearest carbon atom bound with protein oxygen but by carbon indicated in the figure. This can be due to the effect of nitrogen, which increases the charge of its nearest carbon that just transfers the charge of carboxyl oxygen.

Our calculations show that the main role in all functionalization and bonding processes belongs to hybridization and charge transfer. Let us consider the dynamics of hybridization in terms of quantum equations of motion.

For this purpose, let us look into the interaction of drug molecule and functionalized nanotube, which forms new chemical bonds. This process is devoted to the hybridization of separated electrons of drug and carboxyl complex and can be studied in terms of the standard Hamiltonian in the secondary quantization representation (for a single spin subband):(1)$$H=\sum_{l}\varepsilon_{l}c_{l}^{+}c_{l}+\sum_{n}\varepsilon_{n}a_{n}^{+}a_{n}+\sum_{l,n}V_{ln}\left(c_{l}^{+}a_{n}+a_{n}^{+}c_{l}\right),$$
where $c_{l}^{+}{(c}_{l})$ and $a_{n}^{+}(a_{n})$ are the electron creation (annihilation) operators in the Wannier representation for the drug and carboxyl, respectively, $\varepsilon_{l}$ and $\varepsilon_{n}$ are their electron energies measured with respect to the Fermi level, $V_{ln}$ is the hybridization matrix element, and *l* and *n* indicate the atomic positions for the drug molecule and carboxyl, respectively. Using the equation-of-motion method, we can write:(2)$${\dot{c}}_{l}^{+}=i\left[Hc_{l}^{+}\right],\;{\dot{a}}_{n}^{+}=i\left[Ha_{n}^{+}\right]$$
where […] are the commutators.

Simple transformations give the system of equations:(3)$${\ddot{c}}_{l}+\gamma {\dot{c}}_{l}+\sum_{n}{\left|V_{ln}\right|}^{2}c_{l}^{+}=\sum_{n}V_{ln}\omega_{ln}e^{-\omega_{ln}t}a_{n}^{+},$$
(4)$${\ddot{a}}_{n}^{+}+{\sum_{k}\left|V_{kn}\right|}^{2}a_{n}^{+}=\sum_{k}\left(\omega_{kn}-i\gamma \right)e^{\omega_{kn}t}c_{k}^{+},$$
where $\omega_{ln}=\varepsilon_{l}-\varepsilon_{n}$ (in *ħ* units), γ is the decay of the $c_{l}^{+}$ mode due to hybridization. The above cases represent two coupled vibration processes, with *V* and *ω* playing the roles of natural and driving frequencies, respectively.

Seeking a solution to Equation (4) with a two-mode approximation in the form of $a_{n}^{+}=\rho e^{i\omega t}$, we express $a_{n}^{+}$ from Equation (4) and substitute it into Equation (3), which yields:(5)$${\ddot{c}}_{l}^{+}+\gamma {\dot{c}}_{l}^{+}+V^{2}\frac{V^{2}+i\gamma \omega }{V^{2}-\omega^{2}}c_{l}^{+}=0,$$
where $V\equiv V_{ln}$, $\omega \equiv \omega_{ln}$.

This equation describes the reactive (vibrational) and relaxation motions in terms of average values. In both form and meaning, Equation (5) is identical to the phenomenological equation of motion for fluctuations of the “order parameter” [44] in the case-in electron density $c_{l}^{+}$ of the molecule.

Therefore, the coefficient at $c_{l}^{+}$ in Equation (5) is the inverse susceptibility (χ^−1^) *i* in this state of the system. The χ^−1^ value corresponds to the square frequency $\Omega^{2}$ of the dynamic mode of a molecule [44]. If *V* > *ω*, $\Omega^{2}=\frac{V^{4}+i\gamma \omega V^{2}}{V^{2}-\omega^{2}}$, and *V*, *ω* and their ratio determine the tube surface activity with respect to the vibrational motion of the molecule.

This case corresponds to the bond of carboxyl with the tube at *V* > *ω*. As an illustration, the results of calculations of the electron density distribution in carboxyl during hybridization are presented in Figure 8a,c. Such an electron density distribution along with a transferred charge of 0.68 e points to strong hybridization and to a covalent donor–acceptor chemical bond.

However, in addition to the case *V* > *ω*, which corresponds to near-surface molecule locations, states with *ω* > *V* corresponding to drug localization far from the tube surface are possible. In these states, the degree of hybridization and the charge transfer are low. The above can be judged from the electron density distributions in Figure 8b,d.

For these states, the degree of hybridization and the parameter are small (Figure 8b,d) and:(6)$$\Omega =i\frac{V^{2}}{\sqrt{\omega^{2}-V^{2}}}$$

Equation (6) represents a relaxation, diffusion mode in which the motion of the molecule is hydrodynamic and the density fluctuation is $\delta n\sim e^{i(\Omega t-\vec{q}\vec{r})}$ with $\Omega =iDq^{2}$ [45]. Thus, δn~e−1τ and $\frac{1}{\tau }=Dq^{2}$ with *D* being the diffusion coefficient. According to Equation (6), the diffusion coefficient is given by:(7)$$D=\frac{V^{2}}{q^{2}\sqrt{\omega^{2}-V^{2}}}$$

In this state, the molecule behaves in a wave-like diffusion manner, which provides drug delivery from excess-density to deficient-density drug regions at a wavelength $\lambda \sim \frac{2\pi }{q}$ (*q* is the vector near the K point in the Brillion zone of the tube).

The diffusion coefficient *D* depends only on *V* and *ω* and their ratio. If V→ω, the coefficient *D* increases and so do the diffusion and the release capability of the drug attached to carboxyl groups. In diffusion control models [46], the ratio between the amounts of drug $M_{1}$ and $M_{2}$ released at time t is given by:(8)$$\frac{M_{2}}{M_{1}}=\frac{D_{2}^{2}}{D_{1}^{2}}={\left(\frac{V_{2}}{V_{1}}\right)}^{4}\frac{\left(\omega_{1}^{2}-V_{1}^{2}\right)}{\left(\omega_{2}^{2}-V_{2}^{2}\right)},$$
and for ω≫V and $V_{2}∼V_{1}$,
(9)$$M_{2}∼M_{1}\frac{\omega_{1}^{2}}{\omega_{2}^{2}},$$
where $M_{2}$ and $M_{1}$ are the amounts of drug released at *t*_2_ and at *t*_1_, respectively.

Because *ω* depends on the drug concentration (*c*), we can calculate the dependence $M_{2}(c)$ through the substitution of *ω* for different concentrations with $M_{1}$ taken equal to unity (Figure 9c).

Figure 9 also shows the calculated concentration dependences of the binding energy (discussed above); frequencies *ω* for peaks 1 and 2, as well as 3 and 4; and protein release *M*.

For S1 and S2, the values of *ω* and *M* are the same; that is, increasing the oxygen concentration in carboxyl to 4 at. % does not affect the release. During the hybridization of a HIF-1α molecule with carboxyl at 6 at. % of oxygen (S3), there appear to be two bonds (Figure 9c, inset), and hence two frequencies *ω*: $\omega_{12}$ and $\omega_{34}$ (Figure 9b). The frequency $\omega_{12}$ is related to OH–OH, and $\omega_{34}$ is related to OH–O. The frequency $\omega_{12}$ for S3 is much higher than its values for S1 and S2, and the frequency $\omega_{34}$ corresponding to OH–O is much lower than $\omega_{12}$. This provides two possibilities for protein release variation (Figure 9c). The first possibility corresponds to the decrease in *M* with increasing concentration at $\omega_{12}$ (Equation (9)) and the second one to its increase at $\omega_{34}$ (Equation (9)). The first release is possible with small amounts of oxygen, and the second with its excess (in this case, due to oxygen present in the tube: peak 4 in Figure 6c). Because the behavior of molecules and possible release are determined only by the bond of –OH of a HIF-1α molecule with OH and O of carboxyl, the excess of oxygen can also be due to other factors, e.g., to the bond of HIF-1α with O and OH of a cell (hypoxia, oncological diseases, etc.). In this case, the behavior of M determines the regulation mechanism of HIF-1α interaction with oxygen in cells: with a lack of oxygen (OH–OH bond); the release of HIF-1α from a cell remains almost unchanged with increasing concentration, and with its excess (ON–O bond) the release is enhanced. It is this effect that is observed experimentally [18,47].

In summary, carboxyl functionalization decreases the drug tube hybridization by decreasing the natural frequency of molecule vibrations. On the contrary, the driving frequency increases and the drug shows a relaxing, wavelike diffusion behavior as its “central mode” patten. If the natural frequency does not vanish in the diffusion process, low-frequency natural vibrations of drug molecules can occur. Such a phenomenon has been observed time and again [48].

## 3. Materials and Methods

In our study, we used N-MWCNTs synthesized by catalytic chemical vapor deposition (CCVD) in a gas flow reactor at 800 °C for 1 h. The catalyst was nickel nanopowder produced via NiC_2_O_4_ decomposition on a quartz substrate, and the precursor with nitrogen atoms was acetonitrile.

After the preparation of N-MWCNTs, their suspension with HCl was placed in an Eppendorf tube and treated in an ultrasonic water bath for 60 min. Then, they were washed in deionized water until neutral pH was reached, dried in an electric tube furnace under vacuum conditions (10^−1^ Torr), and deposited by aerosol spraying on a stainless steel substrate. All the data for structure and composition of initial N-MWCNTs are shown in Appendix A.

The created layer of N-MWCNTs was irradiated on an ion implanter in an argon atmosphere at an operating pressure of 5 × 10^−4^ Torr, ion bean current of ~100 mA, and average ion energy of 5 keV. The beam diameter in the substrate region was 10 cm^2^, and the beam fluence was 5 × 10^16^ ion/cm^2^.

The structure of irradiated N-MWCNT specimens was analyzed by HRTEM, Raman spectroscopy, XPS and NEXAFS.

The HRTEM images were taken in the bright field mode at an accelerating voltage of 200 kV using a JEM 2200FS (JEOL, Tokyo, Japan) transmission electron microscope and were processed using the Gatan software.

The Raman spectra were measured on a T64000 (HORIBA Jobin Yvon, Longjumeau, France) spectrometer at room temperature; the exciting laser radiation wavelength was 514 nm. The G and D bands were analyzed using the Lorentz functions of the OriginLab software 30.1.109.

The chemical state and the local atomic structure of carbon, oxygen, and nitrogen were analyzed by XPS and NEXAFS on the equipment of the RGL-PES station of the BESSY II electron storage ring (Berlin, Germany). The survey and core-level C1s and N1s PE spectra were recorded at photon energies of 850 and 400 and 500 eV, respectively. The C1s NEXAFS spectra were measured using total electron yield (TEY) mode.

In our theoretical analysis, we used the density functional theory (DFT) and quantum equations of motion. The DFT method in its self-consistent-charge density-functional tight-binding (SCC DFTB) variant [49] was used to model the atomic structures of N-MWCNTs and to analyze their electron-energy properties. In our previous studies, the method was successfully applied to model the functionalization of graphene with oxygen-containing groups and transition metal oxides [50,51]. The total structure energy *E_tot_* was estimated as the sum of the band structure energy *E_BS_*, repulsion energy *E_rep_*, and self-consistent charge energy *E_SCC_*:(10)$$E_{tot}=E_{BS}+E_{rep}+E_{SCC}$$

For estimating the activation energy of binding, the repulsion energy was calculated as:(11)$$E_{rep}=-\frac{1}{2}\int V_{H}\left[n_{0}(r)\right]n_{0}(r)+E_{XC}[n_{0}]+E_{II}-\int V_{XC}\left[n_{0}\right](r)n_{0}(r)$$
where $n_{0}$ is the density comprising free atom densities, $V_{H}$ is the Hartree potential, $V_{XC}$ is the exchange correlation potential, $E_{XC}$ is the exchange correlation energy, and $E_{II}$ is the ion interaction energy.

The interaction of C, N, O, and H atoms was described by the basic set of parameters 3ob-3-1 developed for biological and organic molecules [52]. To identify equilibrium structures, total energy (1) was minimized along the atomic coordinates by the conjugate gradient method at an electron temperature of 300 K, as long as the force on an atom was no greater than 1 × 10^−4^ eV/atom. To calculate the band structure, the reciprocal space was broken by the Monkhorst–Pack method with a grid of 16 × 16 × 1 for COOH-N-MWCNTs and of 8 × 8 × 8 for protein attachment to functionalized N-MWCNTs. The Gauss broadening function with broadening width *σ* = 0.1 was applied to eigenvalues to obtain pDOS plots.

The atomic charge distribution was calculated using the Mulliken method [53], according to which the charge on each atom *Z* is given by:(12)$$Z=Z_{A}-{GAP}_{A}$$
where $Z_{A}$ is the atomic number in the periodic table, and ${GAP}_{A}$ is the total orbital charge for atom *A*.

The binding energy $E_{b}$ between objects *M* and *N* was estimated by the classical formula:(13)$$E_{b}(M+N)=E(M+N)-E(M)-E(N)$$
where E(M) and E(N) are the energies of objects *M* and *N* in isolated states, and *E*(*M* + *N*) is their energy after binding.

## 4. Conclusions

Thus, the results of our research shows that the size of crystallites, topological defects, and nitrogen dopants are significant for carboxyl functionalization. The defects are located mostly at the crystallite interfaces. Under the functionalization, oxygen and OH are bonding with the carbon setting near the defects.

The supercells of N-MWCNT with 2.7, 4, and 6 at. % concentration of oxygen in the carboxyl group were built with the DFTB method according to experimental data. Binding energy and lengths between COOH and tube, and value of the transferred charge into the carboxyl group do not depend on the concentration.

The O and OH components of carboxyl acquire a charge of 0.55 and 0.13 e from a carbon atom after attachment. This points to a substantial energy of hybridization between N-MWCNT and carboxyl. The PDOS peaks of carboxyl and N-MWCNT fall on the same energies, which is an essential condition of covalent bonding.

Our analysis of repulsion energy profiles confirms that the presence of Stones–Wales defects decreases the energy barrier for carboxyl formation due to a decrease in the interaction of carboxyl carbon and its neighbors with the rest of the tube atoms.

The basis for reliable drug delivery is topological defects, functionalization, and wave-diffusion relaxation.

The mechanism of oxygen regulation with HIF-1α molecules is determined by the OH–(OH–C) and OH–(O=C) chemical bonds and driving frequencies of molecule vibrations during hybridization.

Hybridization plays the main role in functionalization, relaxation, oxygen regulation, and possibility of drug delivery. However, for drug delivery, the diffusion equation under certain boundary conditions should be solved.

The concentration dependence of drug release in the diffusion regime shows that the best pattern for drug delivery is provided by the tube with a carboxyl oxygen concentration of 6 at. %.

## Figures and Tables

**Figure 1 ijms-24-01296-f001:**
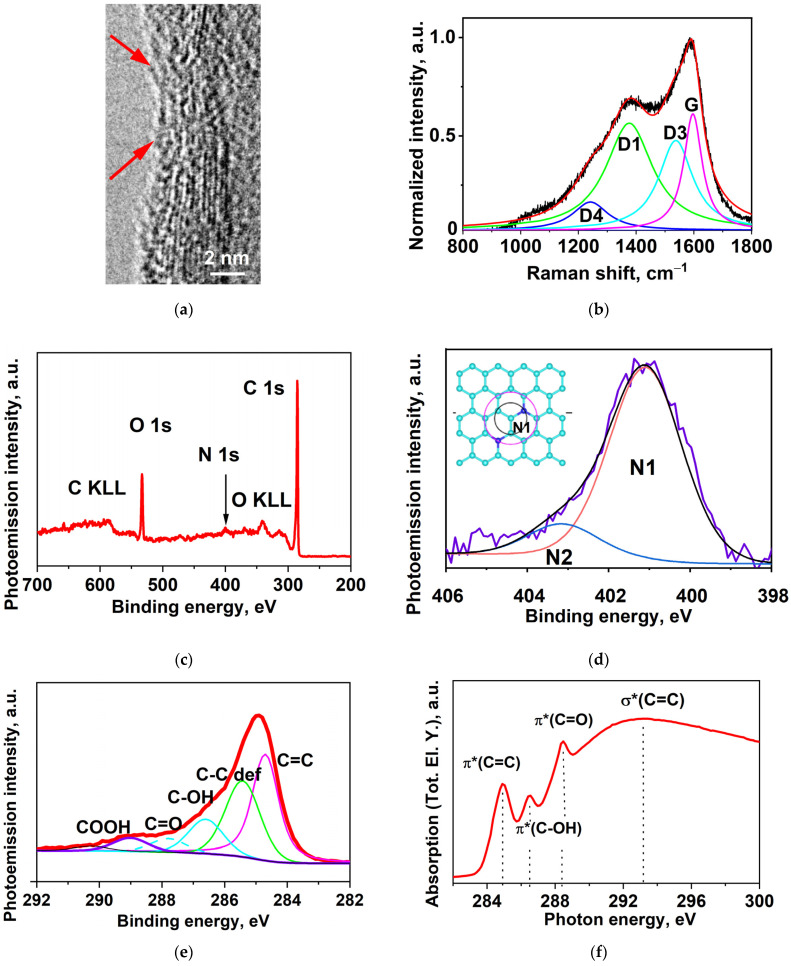
Experimental data on atomic and electronic structure of N-MWCNTs: (**a**) TEM, (**b**) normalized and baseline corrected Raman spectra, (**c**) survey, (**d**) core-level N1s, (**e**) C1s PE spectra of N-MWCNTs, and (**f**) their C1s NEXAFS spectrum.

**Figure 2 ijms-24-01296-f002:**
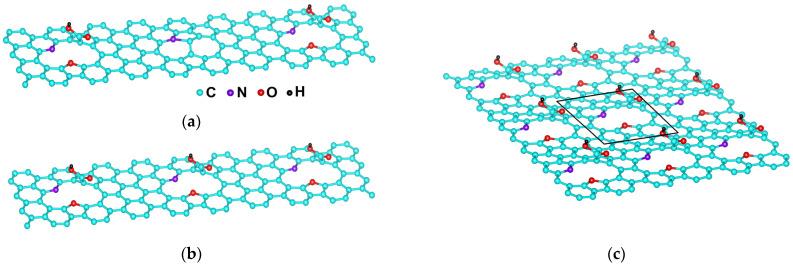
Cells of functionalized N-MWCNTs with (**a**) S1; (**b**) S2; and (**c**) S3 structures.

**Figure 3 ijms-24-01296-f003:**
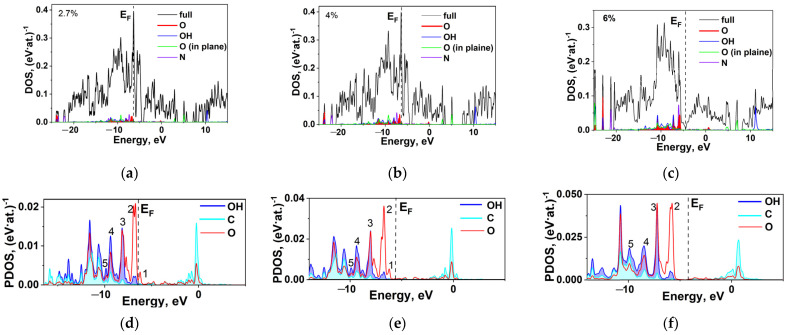
Electronic DOS and PDOS in functionalized N-MWCNTs with (**a**,**d**) S1; (**b**,**e**) S2; and (**c**,**f**) S3 structure.

**Figure 4 ijms-24-01296-f004:**
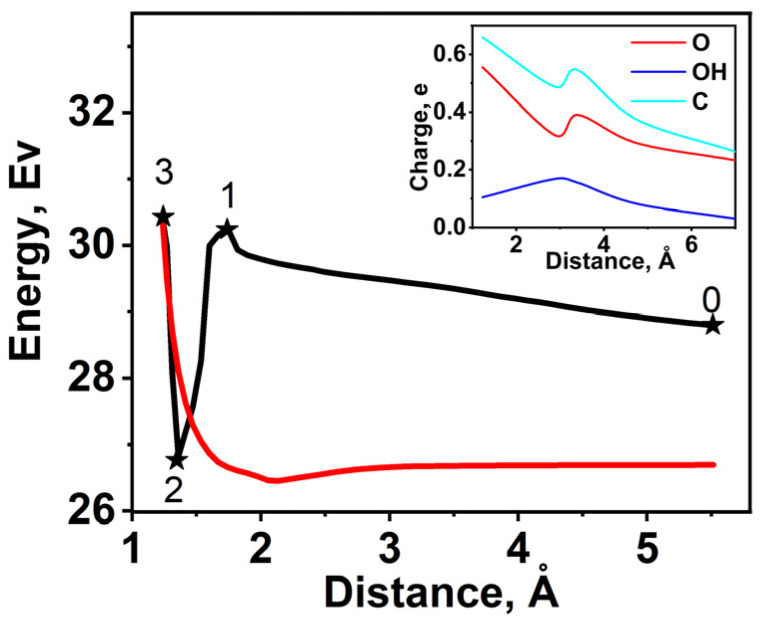
Repulsion energy of OOH group and tube surface *E_rep_* versus interatomic distance C=O with red curve for rigidly fixed C atom in surface layer, black curve for loose C atom, and inset for charge transfer on C, O, and OH versus interatomic distance C=O after relaxation for loose C.

**Figure 5 ijms-24-01296-f005:**
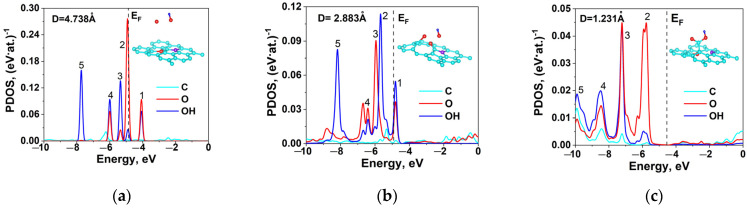
PDOS of carboxyl components C, O, and OH for three different distances in bond C=O with inset showing respective spatial positions of O and OH relative to tube surface: (**a**) 4.738 Å; (**b**) 2.883 Å; (**c**) 1.231 Å.

**Figure 6 ijms-24-01296-f006:**
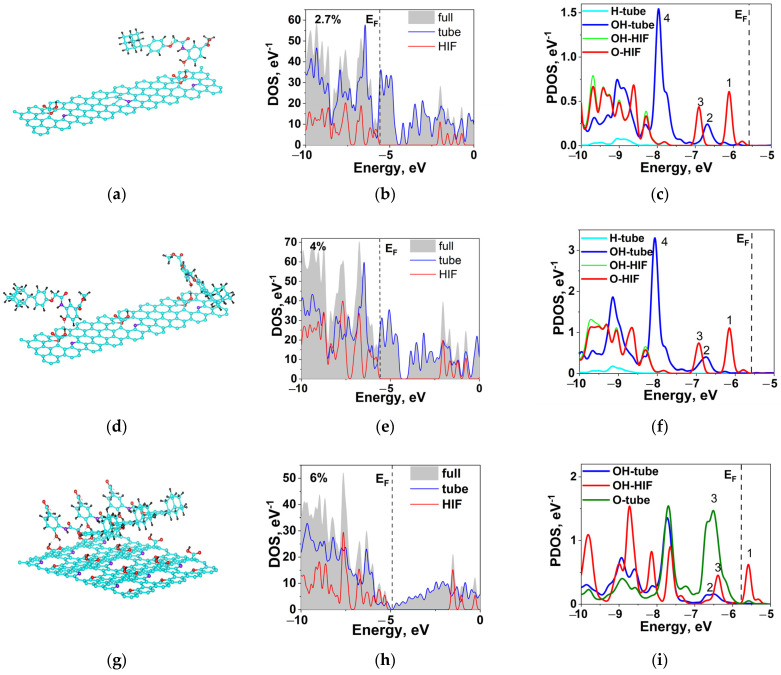
HIF-1α-COOH-N-MWCNT with S1, S2, and S3: (**a**–**c**) Atomic structure; (**d**–**f**), Electronic DOS, and (**g**–**i**) PDOS.

**Figure 7 ijms-24-01296-f007:**
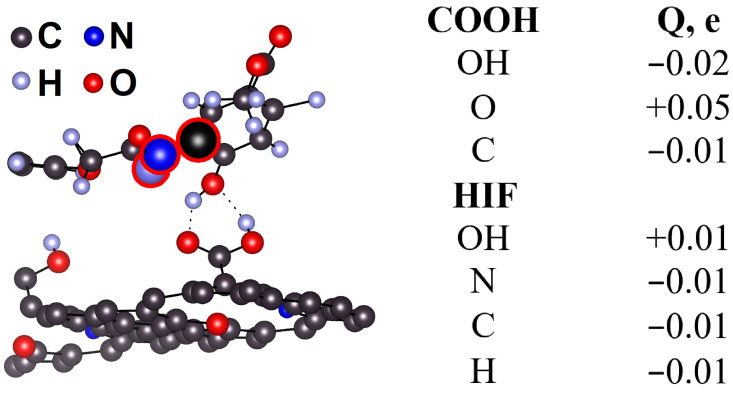
Transferred charge *q* between carboxyl and HIF-1α.

**Figure 8 ijms-24-01296-f008:**
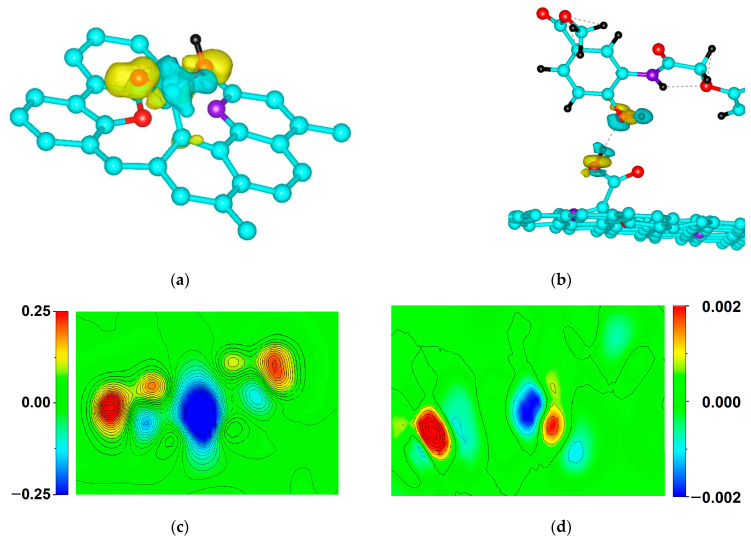
Comparison of electron density distribution between carboxyl and HIF-1α molecule: (**a**) Spatial location of carboxyl; (**b**) Spatial location of and HIF-1α molecule; (**c**) Electron density distributions in carboxyl and tube; (**d**) Electron density distributions in HIF-1α molecule and carboxyl.

**Figure 9 ijms-24-01296-f009:**
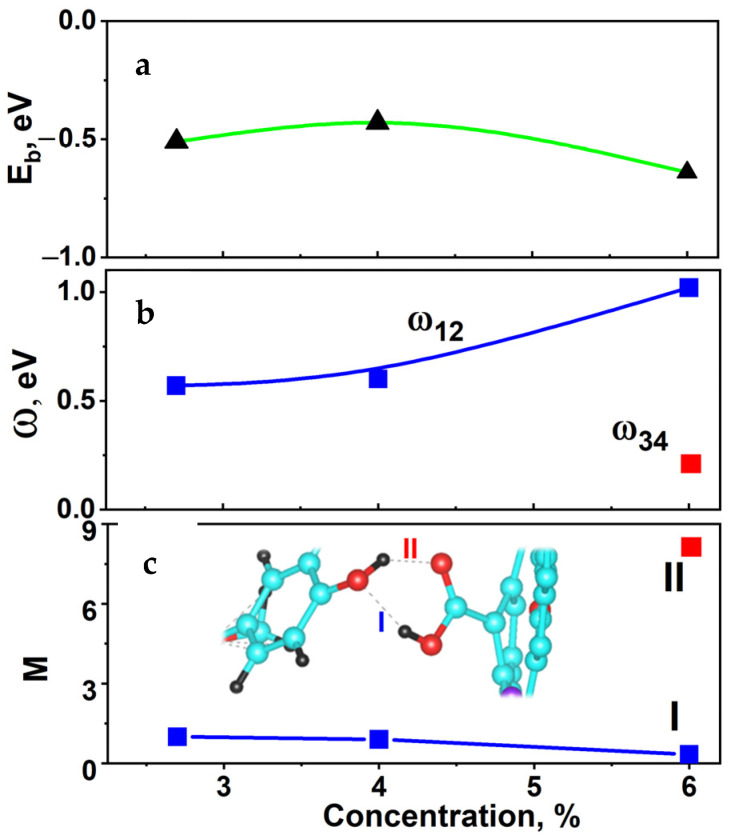
Concentration dependences of binding energy (**a**) E_b_, (**b**) driving frequencies, (**c**) and HIF-1α release *M*.

**Table 1 ijms-24-01296-t001:** Results of the Raman spectra analysis.

G, cm^−1^	D1, cm^−1^	D3, cm^−1^	D4, cm^−1^	I_D_/I_G_	I_D1_/I_D_	I_D3_/I_D_	I_D4_/I_D_
1597	1385	1539	1242	3.0	0.7	0.2	0.1

**Table 2 ijms-24-01296-t002:** Electronic characteristics of functionalized N-MWCNTs.

Characteristic	S1	S2	S3
*E_F_*, eV	−5.58	−5.60	−4.48
n(*E_F_*), (eV·atom)^−1^	0.25	0.25	0.01
*E_b_*, eV	−8.18	−7.91	−8.28
R (C=O), Å	1.23		
R (C–OH), Å	1.36		
q(C), (e/atom)	−0.68		
q(O), e	+0.55		
q(OH), e	+0.11		
q(C), e	+0.02		

**Table 3 ijms-24-01296-t003:** Main PDOS parameters of carboxyl components C, O, and OH.

R, Å	E_F_, eV	q(C), e/Atom	q(O), e/Atom	q(OH), e/Atom	Peak 1, eV	Peak 2, eV	Peak 3, eV	Peak 4, eV	Peak 5, eV
4.738	−0.8667	0.014	0.056	0.072	−4.08	−4.92	−5.35	−6	−7.73
2.883	−0.9435	0.012	0.061	0.073	−4.81	−5.66	−5.96	−6.70	−8.18
1.231	−0.4831	0.012	0.053	0.04	-	−5.77	−7.22	−8.47	−9.90

**Table 4 ijms-24-01296-t004:** Positions of PDOS peaks for HIF-1α and carboxyl.

Peak, eV Concentration	1	2	3	4	Ω
S1	−6.11 (HIF-O)	−6.68 (tube-OH)	−6.89 (HIF-O)	−7.98 (tube-OH)	0.57 (1–2)
S2	−6.15 (HIF-O)	−6.75 (tube-OH)	−6.94 (HIF-O)	−8.06 (tube-OH)	0.60 (1–2)
S3	−5.58 (HIF-OH)	−6.60 (tube-OH)	−6.39 (HIF-OH)	−6.59 (tube-O)	1.02 (1–2) 0.20 (3–4)

**Table 5 ijms-24-01296-t005:** Electronic characteristics of HIF-COOH-N-MWCNT structure.

Characteristic	S1	S2	S3
*E_F_*, eV	−5.56	−5.60	−4.88
n(*E_F_*), (eV·atom)^−1^	0.168	0.092	0.001
*E_b_*, eV	−0.51	−0.51/−0.35	−0.64
R (OH-OH), Å	1.60	1.60/1.90	1.88
R (O-OH), Å	-	-	1.94

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
