# Peer review of "Carboxyl Functionalization of N-MWCNTs with Stone–Wales Defects and Possibility of HIF-1α Wave-Diffusive Delivery"

_ijms, 2023, doi:10.3390/ijms24021296_

Round 1
Reviewer 1 Report
The undoubted advantages of the article are its completeness and the variety of experimental and theoretical approaches used. This makes the article interesting for readers and potentially highly cited. However, a few points should be improved before the accepting of the article.
1) Why the authors considered only two chemical states for nitrogen atoms after ion beam irradiation?
2) There is the lack of experimental information about nanotubes before irradiation. Such information is very suitable for correct evaluation of changes appearing after irradiation.
3) What is the nitrogen concentrations before and after irradiation? This data should be added.
4) If the authors' DFT approach provides the energies of 1S states, it would be interesting to compare the energies of these states with the experimentally measured peaks shown in Figs. 1c and 1d.
5) If the PDOS in Figure 3 were subjected to any processing (Gaussian/Lorentz broadening, etc.), this should be described in the figure caption. This is important for reproducibility of the presented results.
Author Response
We thank Reviewer for his kind revision. We prepared Supplementary.doc that content required data and expanded text of the article. The added text is highlighted. Here are answers on Reviewer's notes
1) Why the authors considered only two chemical states for nitrogen atoms after ion beam irradiation?
Initial N-SWCNT contain four types of nitrogen configurations and molecular nitrogen in nanotubes walls. Irradiation destructs nanotubes walls that leads to elimination of molecular nitrogen for a space between walls and to failure of the less stable nitrogen configuration. So irradiated SWCNT contain nitrogen in only two configurations.
2) There is the lack of experimental information about nanotubes before irradiation. Such information is very suitable for correct evaluation of changes appearing after irradiation.
All the data for structure and composition of initial N-SWCNT is shown Supplementary Figures S1 and S2.
The corresponding text is added to «Materials and Methods».
3) What is the nitrogen concentrations before and after irradiation? This data should be added.
Nitrogen concentration in initial N-MWCNTs is about 4at%, after irradiation this value decreases to about 2at% (See Figure S1 in Supplementary).
4) If the authors' DFT approach provides the energies of 1S states, it would be interesting to compare the energies of these states with the experimentally measured peaks shown in Figs. 1c and 1d.
In the current paper we applied DFTB approach that doesn’t provide calculation of photoemission intensity in dependence on binding energy. It’s a complex task that wasn’t the goal of this article.
5) If the PDOS in Figure 3 were subjected to any processing (Gaussian/Lorentz broadening, etc.), this should be described in the figure caption. This is important for reproducibility of the presented results.
The Gauss broadening function with broadening width σ=0.1 was applied to eigenvalues to obtain pDOS plots. This text is added to Materials and Methods.

Reviewer 2 Report
The manuscript of “Carboxyl Functionalization of N-MWCNTs with Stone-Wales 2 Defects and Possibility of HIF-1α Wave-diffusive Delivery” reports a nitrogen-doped multi-walled carbon nanotubes catalytic chemical vapor deposition and irradiated with argon ions. HIF-1α molecules were incorporated to nanotubes by OH or COOH groups. The mechanism of hybridization and relaxation were demonstrated systematically. So I think this article will attract wide audience, and I recommend accept this manuscript after minor revisions.
(1) In Figures, there are a, b, c and so on in captions, but there are A, B, and C in Figures, it is better to unify them. And in Figure 3, there are no captions for d-f. Part of Figures in main article is full name, and part of Figures in main article is abbreviation as Fig.X, for example Fig.2a, b in line 248. Please refer to “Opt. Mater. 2023, 135, 113288”,“Opt. Mater. 2023,135, 113219.” and “Nanomaterials 2020, 10(5), 944” for normal format, please cite the above three references.
(2) In Table 3 and 5, the significant numbers should be unified for every columns.
Author Response
We thank Reviewer for his work. Here are answers on his notes.
Answers to Reviewer 2.
- In Figures, there are a, b, c and so on in captions, but there are A, B, and C in Figures, it is better to unify them. And in Figure 3, there are no captions for d-f. Part of Figures in main article is full name, and part of Figures in main article is abbreviation as Fig.X, for example Fig.2a, b in line 248.
Symbols in captions and Figures are unified. Captions for d-f of Figure 3 are added. All Figures are full-named in the text.
- Please refer to “Opt. Mater. 2023, 135, 113288”,“Opt. Mater. 2023,135, 113219.” and “Nanomaterials 2020, 10(5), 944” for normal format, please cite the above three references.
We read these papers but they are devoted to solar cells that are unfortunately too far from our object of study. So we can’t cite these papers. We thank Reviewer for interesting paper and we’ll try to cite them in future.
- In Table 3 and 5, the significant numbers should be unified for every columns.
It is corrected now.